# Depression and anxiety in people with cognitive impairment and dementia during the COVID-19 pandemic: Analysis of the English Longitudinal Study of Ageing

**Brian Beach**[1]*, **Andrew Steptoe**[2], **Paola Zaninotto**[1]

**1** Research Department of Epidemiology & Public Health, Institute of Epidemiology & Health Care, University College London, London, United Kingdom, **2** Research Department of Behavioural Science & Health, Institute of Epidemiology & Health Care, University College London, London, United Kingdom

* b.beach@ucl.ac.uk

**Data Availability Statement:** This study uses data from the English Longitudinal Study of Ageing, which are freely available upon registration through the UK Data Service (https://doi.org/10.5255/

## Abstract

### Background

Some studies have identified declines in mental health during the Coronavirus Disease 2019 (COVID-19) pandemic in different age groups, including older people. As anxiety and depression are common neuropsychiatric symptoms among people with cognitive impairment, the mental health experiences of older people during the pandemic should take cognitive function into consideration, along with assessments made prior to the pandemic. This study addresses evidence gaps to test whether changes in depression and anxiety among older people through the COVID-19 pandemic were associated with cognitive impairment. It also investigates whether associations varied according to key sources of sociodemographic inequality.

### Methods and findings

Using data from the English Longitudinal Study of Ageing (ELSA) collected from 2018/2019 to November/December 2020, we estimated changes in depression and anxiety for people aged 50+ in England across 3 cognitive function groups: no impairment, mild cognitive impairment, and dementia. Conditional growth curve models were estimated for continuous measures over 3 time points ($N = 5,286$), with mixed-effects logistic regression used for binary measures. All models adjusted for demographics (age, gender, ethnicity, and cohabiting partnership), socioeconomics (education, wealth, and employment status), geography (urban/rural and English region), and health (self-rated and the presence of multimorbidity).

We found that depression (measured with CES-D score) worsened from 2018/2019 to November/December 2020 for people with mild cognitive impairment (1.39 (95% CI: 1.29 to 1.49) to 2.16 (2.02 to 2.30)) or no impairment (1.17 (95%CI: 1.12 to 1.22) to 2.03 (1.96 to 2.10)). Anxiety, using a single-item rating of 0 to 10 also worsened among those with mild cognitive impairment (2.48 (2.30 to 2.66) to 3.14 (2.95 to 3.33)) or no impairment (2.20 (2.11 to 2.28) to 2.85 (2.77 to 2.95)). No statistically significant increases were found for those

UKDA-SN-5050-24, https://doi.org/10.5255/UKDA-SN-8688-3).

**Funding:** The English Longitudinal Study of Ageing was developed by a team of researchers based at University College London, NatCen Social Research, the Institute for Fiscal Studies, the University of Manchester and the University of East Anglia. The work for this paper was supported by funding from the National Institute for Health and Care Research (NIHR) [NIHR202472] (PI: PZ). ELSA is funded by the National Institute on Aging (R01AG017644) and by UK Government Departments coordinated by the National Institute for Health and Care Research (NIHR). Data collection during the COVID-19 pandemic was supported by the Economic and Social Research Council (ES/V003941/1) (PI: AS). The funders had no role in study design, data collection and analysis, decision to publish, or preparation of the manuscript.

**Competing interests:** The authors have declared that no competing interests exist.

**Abbreviations:** BPSD, behavioral and psychological symptoms of dementia; CES-D, Center for Epidemiological Studies Depression; COVID-19, Coronavirus Disease 2019; ELSA, English Longitudinal Study of Ageing; GAD-7, 7-item Generalized Anxiety Disorder; GHQ-12, 12-item General Health Questionnaire; HCAP, Harmonised Cognitive Assessment Protocol; MCAR, missing completely at random; MCID, minimal clinically important difference; MICE, multiple imputation by chained equations.

with dementia. Using a clinical cutoff for likely depression (CES-D $\geq$4), we found statistically significant increases in the probability of depression between 2018/2019 and November/December 2020 for those with no impairment (0.110 (0.099 to 0.120) to 0.206 (0.191 to 0.222)) and mild impairment (0.139 (0.120 to 0.159) to 0.234 (0.204 to 0.263)).

We also found that differences according to cognitive function that existed before the pandemic were no longer present by June/July 2020, and there were no statistically significant differences in depression or anxiety among cognitive groups in November/December 2020. Wealth and education appeared to be stronger drivers for depression and anxiety, respectively, than cognitive impairment. For example, those with no impairment in the richest two-thirds scored 1.76 (1.69 to 1.82) for depression in June/July, compared to 2.01 (1.91 to 2.12) for those with no impairment in the poorest third and 2.03 (1.87 to 2.19) for those with impairment in the poorest third. Results may be limited by the small number of people with dementia and are generalizable only to people living in the community, not to those in institutional care settings.

## Conclusions

Our findings suggest a convergence in mental health across cognitive function groups during the pandemic. This suggests mental health services will need to meet an increased demand from older adults, especially those not living with cognitive impairment. Further, with little significant change among those with dementia, their existing need for support will remain; policymakers and care practitioners should ensure this group continues to have equitable access to mental health support.

## Author summary

### Why was this study done?

- Early research conducted after the start of the COVID-19 pandemic suggested that the pandemic was having a negative impact on mental health.

- Older people with cognitive impairment or dementia are more vulnerable to the negative impacts of the pandemic, and they tend to have worse mental health than older people with no cognitive impairment.

- This study was done to test whether changes in mental health over time through the pandemic was associated with cognitive impairment, along with whether associations varied according to key sources of sociodemographic inequality.

### What did the researchers do and find?

- This study draws on the richness of the English Longitudinal Study of Ageing, a study of people aged 50+ in England, which provides a robust way of assessing cognitive function and mental health (in terms of depression and anxiety) and includes measurements

before the pandemic (2018/2019) and at 2 time points during it (June/July and November/December 2020).

- Using a statistical approach called conditional growth curve modelling, we found that depression and anxiety worsened for people with no cognitive impairment or mild cognitive impairment between 2018/2019 and November/December 2020. Average depression scores increased from 1.17 to 2.03 and 1.30 to 2.16, respectively, while anxiety ratings increased from 2.20 to 2.85 and 2.48 to 3.14.

- When using a measure for likely clinical depression, we found the probability of clinical depression also increase for people with no cognitive impairment or mild cognitive impairment between 2018/2019 and November/December 2020, from 0.110 to 0.206 and 0.139 to 0.234, respectively.

- In terms of inequalities, wealth and education appeared to be stronger drivers for depression and anxiety, respectively, than cognitive impairment. For example, those with no impairment in the richest two-thirds scored 1.76 for depression in June/July, compared to 2.01 for those with no impairment in the poorest third and 2.03 for those with impairment in the poorest third.

## What do these findings mean?

- Our findings suggest a convergence in mental health over time among different cognitive function groups, with similar outcomes in November/December 2020 for those with no impairment, mild cognitive impairment, or dementia.

- Health professionals who provide mental health support to older people in the community should be aware that increasing demand for support is likely to come from those with no or mild cognitive impairment.

- With little significant change in mental health for those with dementia, those providing support will need to ensure this group continues to access services despite competing demands from those with no or mild cognitive impairment.

## Introduction

Researchers and policymakers continue to be interested in the impact of the Coronavirus Disease 2019 (COVID-19) pandemic on mental health. Studies have identified declines in mental health over the course of the pandemic across the world, linked to concerns over infection, the consequences of lockdown and isolation measures, risks related to job insecurity and financial worries, and disruption in day-to-day activities [1–5]. While early findings drew on internet-based surveys during the pandemic [6–8], more recent research has examined pandemic experiences compared to information collected before the pandemic [2,9–15].

Different groups of the population face distinct challenges with respect to maintaining good mental health and how the pandemic impacted their lives. The prevalence of anxiety or depression has been found to decline with increasing age [16,17]. While older adults are at greater risk of adverse outcomes from exposure to COVID-19 and have been linked to greater

worries about it [18], older age has been linked to better mental health in terms of anxiety and depression during the pandemic [19,20]. Nonetheless, older people did experience a deterioration in mental health over the course of the pandemic compared to before its onset [21].

Older adults are also not a homogeneous group, and the likelihood of living with conditions such as cognitive impairment or dementia increases with age. Around 6.7% of people aged 65+ in England were estimated to have dementia in 2015 [22]. For mild cognitive impairment, estimates for prevalence range from 5.0% to 36.7% depending on the various definitions and diagnostic criteria used in different studies [23]. Moreover, anxiety and depression are common neuropsychiatric symptoms among people with dementia or mild cognitive impairment [24–26]. One meta-analysis estimated the pooled prevalence of depression and anxiety among people with dementia at 39% each [26]; this compares to estimated prevalence rates of 13.3% for depression in the overall older population and between 1.2% and 15% for anxiety in community samples of older people [27,28]. Prevalence rates further vary according to age, for example, with estimates for depression of 17.1% among those aged 75+ and 30% to 50% for those aged 90+ [29]. Such neuropsychiatric symptoms may also predict conversion from mild cognitive impairment to dementia [30], although the evidence is mixed for anxiety and depression specifically [31,32].

Examining the mental health experiences of older people during the pandemic should, therefore, take cognitive function into consideration. Some research has already investigated this to an extent; a rapid review of evidence related to the impact of COVID-19 isolation measures on mental health among people with dementia found that most studies identified worsening behavioral and psychological symptoms of dementia (BPSD) [33]. Some of the research found these results through subjective assessments made by caregivers [34], qualitative interviews of people with dementia and their caregivers [35,36], and across international contexts [37].

Only limited evidence has examined mental health among people with dementia across the pandemic using quantitative measures that were also assessed prior to the pandemic [38]; quantitative longitudinal research on the impact of COVID-19 on people with dementia was a key direction for future research called for by the United Kingdom–based expert working group on dementia well-being and COVID-19 [39]. Most of the existing relevant research on cognitive function and mental health during the pandemic also does not differentiate between dementia and mild cognitive impairment, although one small study from Greece did examine this distinction with respect to pre-pandemic measures [40].

This study addresses these gaps in the evidence base by testing whether changes in depression and anxiety among older people during the COVID-19 were associated with cognitive impairment. Using data from the English Longitudinal Study of Ageing (ELSA), we examine changes in depression and anxiety from before the pandemic (2018/2019) across 2 time points during the pandemic (June/July and November/December 2020) with respect to 3 levels of cognitive function. We also investigate whether the associations between cognitive function and mental health varied according to key sources of sociodemographic inequalities related to wealth, education, geographic region, and multimorbidity.

## Methods

### Data

Our project used data collected before and throughout the COVID-19 pandemic as part of the ELSA [41]. ELSA follows a representative sample of people aged 50+ across England since 2002, covering topics such as health, finances, and psychosocial well-being, with refreshment samples added periodically to ensure representativeness over time.

This analysis draws on the COVID-19 sub-study conducted as part of ELSA in 2020 [42]. ELSA members and their partners participated in 2 special surveys conducted in June/July and November/December 2020, capturing their perspectives during the pandemic. Response rates were notably high, at 75% for each wave of data collection and 94% longitudinally. For measures prior to the pandemic, we draw on pre-pandemic responses to the main ELSA survey (Wave 9), collected in 2018/2019. As our study made exclusive use of secondary data analysis, ethics approval was not applicable for the work presented here.

## Measures

Our primary outcomes of interest include measures for depression and anxiety. Depression was measured using the 8-item Center for Epidemiological Studies Depression (CES-D) scale, a validated and reliable instrument for assessing depression among older adults [43]. The scale draws on responses to 8 yes/no questions to provide a continuous measure ranging 0 to 8 with higher scores reflecting greater levels of depressive symptoms. A binary measure was also constructed where scores of four or more were used to identify likely cases of clinical depression [44].

Anxiety was assessed using the 7-item Generalized Anxiety Disorder (GAD-7) scale, which has demonstrated validity and reliability for screening generalized anxiety disorder and assess its severity [45]. Each item is measured on a 4-point Likert scale ranging 0 to 3, providing GAD-7 scores ranging 0 to 21 with higher scores reflecting greater severity and association with higher levels of functional impairment. A binary measure to assess cases of generalized anxiety disorder was constructed using scores of 10 or more. The GAD-7 scale was only measured during the ELSA COVID-19 sub-study, restricting analyses to the 2 time periods included there.

Additional analyses also examined anxiety using a single-item response measured 0 to 10 that was included in ELSA Wave 9 as well as the COVID-19 sub-study, providing assessments at 3 time points. Some studies suggest that there is similar sensitivity and specificity between such single items and multiple-item scales for anxiety [46–48]. Using this measure will provide some insight into changes from before the pandemic.

The main exposure in our analysis is cognitive function status, classified as no cognitive impairment, mild cognitive impairment, or dementia. Individuals' classification draws on work from another ELSA sub-study from 2018, the Harmonised Cognitive Assessment Protocol (HCAP). ELSA-HCAP applied a range of questionnaires and other evaluations used in clinical and nonclinical settings to assess participants' cognition function [49]. From this work, a predictive algorithm was developed to classify all ELSA respondents aged 60+ into one of 3 cognitive function groups: no impairment, mild impairment, or dementia [50].

## Analytical approach

Our analytical approach was planned during the conception of the project and no data-driven changes to this plan took place. Given existing knowledge about the link between mental health and cognitive impairment, along with early research on the impact of the pandemic on mental health, our analyses tested the hypotheses that changes in mental health, in terms of depression and anxiety, was associated with cognitive impairment over time from before to during the COVID-19 pandemic. We also investigated whether these associations varied according to key sources of sociodemographic inequality in England, i.e., education, wealth, geographic region, and multimorbidity.

For the continuous measure of depression, we estimated a conditional growth curve model to assess change in depression score by cognitive function status, employing maximum

likelihood estimation with unstructured covariance. This approach was also taken for the single-item measure of anxiety available at 3 time points. Given we only have 2 time points when GAD-7 was measured, we used a population-averaged fixed-effects model with robust standard errors to assess associations with cognitive function. With respect to the binary measures reflecting likely cases of clinical depression or generalized anxiety disorder, we applied mixed-effects logistic regression with independent variance for the random effect of time.

Models controlled for pre-pandemic (2018 to 2019) measures covering demographics (age, gender, ethnicity, and cohabiting partnership status), socioeconomics (education, wealth, and employment status), geography (urban/rural and English region), and health (self-rated health and the presence of multimorbidity). Multimorbidity was classed as the presence of two or more of the following diagnosed conditions: high blood pressure/hypertension; angina or heart attack; congestive heart failure; diabetes; stroke; chronic lung disease or asthma; cancer; and dementia, senility, serious memory impairment, or Alzheimer's disease. For the continuous CES-D and GAD-7 scales, we additionally tested for systematic inequalities by introducing three-way interactions among cognitive impairment (a binary measure combining the mild impairment and dementia groups), measurement wave, and binary measures for education, wealth, region, and multimorbidity in separate models, using the same control variables listed above.

Missingness on single items can introduce bias when using multiple-response scales such as CES-D and GAD-7. In addition, a survey error during the first COVID-19 survey resulted in the eighth depression item not being asked to around 75% of respondents. These missing values were replaced using 1 cycle of multiple imputation by chained equations (MICE), adjusting for age and gender, given these items can be assumed to be missing completely at random (MCAR) [51,52]. Following this, more than 97% of respondents at any wave were complete on the items for CES-D or GAD-7; MICE was again applied for respondents missing 1 or 2 items in each scale, replacing missing values before generating the summary scores. This raised coverage to over 99.5% across the COVID sub-study waves. Analyses were conducted using Stata 17.0 [53]. This study followed the Strengthening the Reporting of Observational Studies in Epidemiology (STROBE) reporting guideline (S1 STROBE Checklist).

## Results

Table 1 provides descriptive statistics of our analytical sample for depression within the first assessment of the COVID-19 sub-study. The percentages provided reflect similar statistics for other time periods and outcomes of interest.

The analytical samples vary slightly by the outcome of interest, with 5,286 individuals included in analyses for depression and the single-item anxiety measure and 5,281 for the GAD-7 anxiety score. All figures reported below reflect the full model, adjusted for all covariates identified in the previous section, and vertical scales have been restricted to relevant outputs ranges to facilitate visual inspection.

We first present results showing estimated depression scores across the 3 measured time points for the 3 categories of cognitive function (Fig 1). We find that the estimated depression score was significantly different across the 3 cognitive function groups prior to the pandemic. The score for those with no impairment was 1.17 (95%CI: 1.12 to 1.22) compared to 1.39 (1.29 to 1.49) for those with mild impairment and 1.81 (1.53 to 2.10) for those with dementia. Scores increased over time through the pandemic, with statistically significant increases between June/July 2020 and November/December for those with no impairment, going from 1.84 (1.79 to 1.90) to 2.03 (1.96 to 2.10), and those with mild impairment (from 1.89 (1.77 to 2.00) to 2.16 (2.02 to 2.30)). There was no statistically significant change in the score among those with

**Table 1. Sociodemographic characteristics among the sample for depression at the first assessment of the ELSA COVID-19 sub-study (June/July 2020).**

| Characteristic | Percentage | N (of 5,107) |
|---|---|---|
| Cognitive impairment (No impairment) | 77.9 | 3,978 |
| (Mild impairment) | 20.0 | 1,019 |
| (Dementia) | 2.2 | 110 |
| Age (*mean*) | 72.7 | 5,107 |
| Female | 55.7 | 2,843 |
| Non-white | 3.0 | 153 |
| Partner in household | 68.4 | 3,491 |
| Education (high, i.e., degree or equivalent) | 22.5 | 1,148 |
| (medium, i.e., A or O level equiv.) | 45.8 | 2,340 |
| (low, i.e., no qualifications) | 31.7 | 1,619 |
| Employment status (in work) | 15.3 | 782 |
| (retired) | 80.6 | 4,116 |
| (other) | 4.1 | 209 |
| Net wealth (Poorest third) | 33.0 | 1,685 |
| (Middle third) | 33.3 | 1,700 |
| (Richest third) | 33.7 | 1,722 |
| Rural residence | 28.0 | 1,430 |
| Region (The North) | 27.4 | 1,399 |
| (The Midlands) | 21.5 | 1,096 |
| (London and East) | 21.4 | 1,093 |
| (The South) | 29.7 | 1,519 |
| Self-rated health (Excellent or very good) | 41.1 | 2,100 |
| (Good) | 35.7 | 1.825 |
| (Fair or poor) | 23.1 | 1,182 |
| Multimorbidity (2+ health conditions) | 25.1 | 1,282 |

dementia. In June/July and November/December, the differences among cognitive function groups were also no longer statistically significant.

Turning to the anxiety score measured using the GAD-7 scale, Fig 2 shows that the estimated score for people with dementia was higher just after the start of the pandemic in June/July 2020 than for the other cognitive function groups, at 4.59 (3.66 to 5.52) compared to 2.99 (2.72 to 3.26) for those with mild impairment and 2.79 (2.67 to 2.90) for those with no impairment. There was no statistically different change for people with dementia during the pandemic, but the estimated average score did rise by November/December for the mild impairment and no impairment groups, reaching 3.57 (3.28 to 3.86) and 3.02 (2.90 to 3.14), respectively, from 2.99 (2.72 to 3.26) and 2.79 (2.67 to 2.90).

Although the single-item measure of anxiety is different from the GAD-7 scale, it provides added insight here with respect to differences in anxiety before and during the pandemic. Like with GAD-7, we see in Fig 3 a statistically significant increase in the average estimated rating between June/July and November/December for those with no impairment, going from 2.65 (2.57 to 2.74) to 2.85 (2.77 to 2.95). Both the no impairment and mild impairment groups demonstrated a significant increase in anxiety rating between 2018/2019 and November/December 2020; the score for those with no impairment increased from 2.20 (2.11 to 2.28) to 2.85 (2.77 to 2.95), while for the mild impairment group, it went from 2.48 (2.30 to 2.66) to 3.14 (2.95 to 3.33). The apparent change for those with dementia was similar to the other groups, but wide confidence intervals yield no statistically significant differences over time.

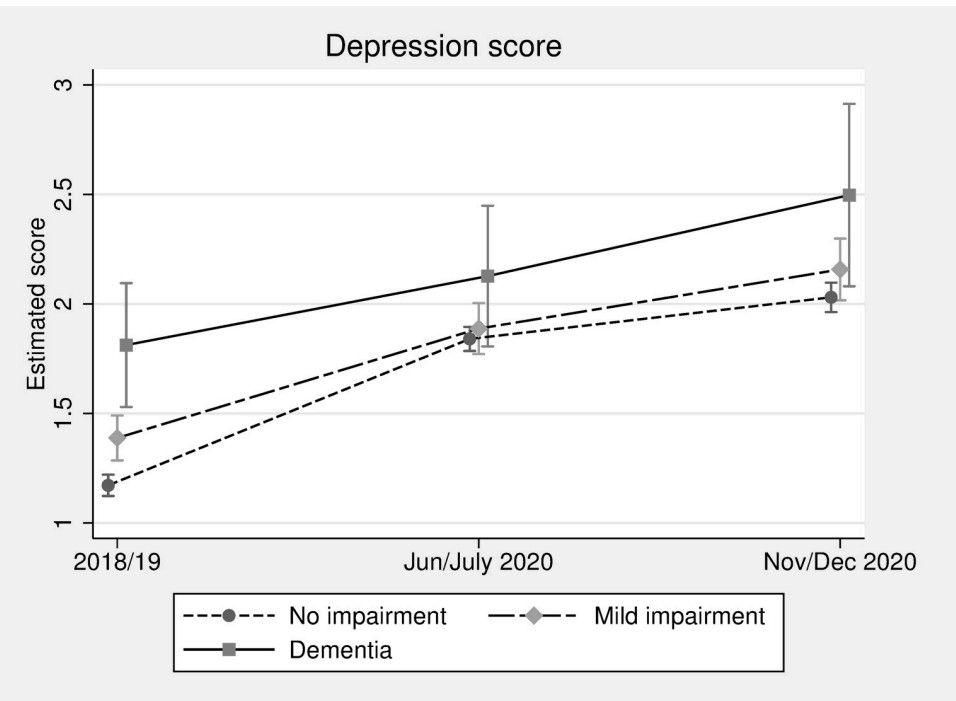

**Fig 1. Depression score (CES-D) over time by cognitive function (estimated scores and 95% confidence intervals, adjusted for all available covariates).**

While the CES-D and GAD-7 scales allow us to examine changes in the average estimated scores, changes in scores alone cannot inform whether these are clinically significant changes or more related to the general unease most people faced due to the unprecedented and unpredictable nature of the pandemic. To explore this, we estimated results based on clinically recognized cutoff values for the 2 scales (Fig 4).

With respect to anxiety, we find no statistically significant differences over time in the probability of likely generalized anxiety disorder as measured using GAD-7, although point estimates suggest a possible decline in probability for those with dementia from 0.165 (0.108 to 0.223) to 0.080 (0.035 to 0.126).

Regarding depression, we find statistically significant increases in the probability of likely clinical depression for those with no impairment and those with mild impairment, looking from before the pandemic to during it. The probability for those with mild impairment was higher than for those with no impairment before the pandemic, at 0.139 (0.120 to 0.159) compared to 0.110 (0.099 to 0.120). These increased significantly after the start of the pandemic by June/July, then reaching 0.234 (0.204 to 0.263) and 0.206 (0.191 to 0.222) by November/December 2020. While there was an increasing trajectory in the point estimates for those with dementia, the differences in estimated probabilities were not statistically significant.

## Inequalities in mental health and cognitive function

To test for systematic inequalities in the association between cognitive function and mental health, we first constructed models controlling only for age and gender. We found no significant three-way interactions, indicating the rate of change in our outcomes was not significantly distinct for people with cognitive impairment across distinct social, economic, and health groups.

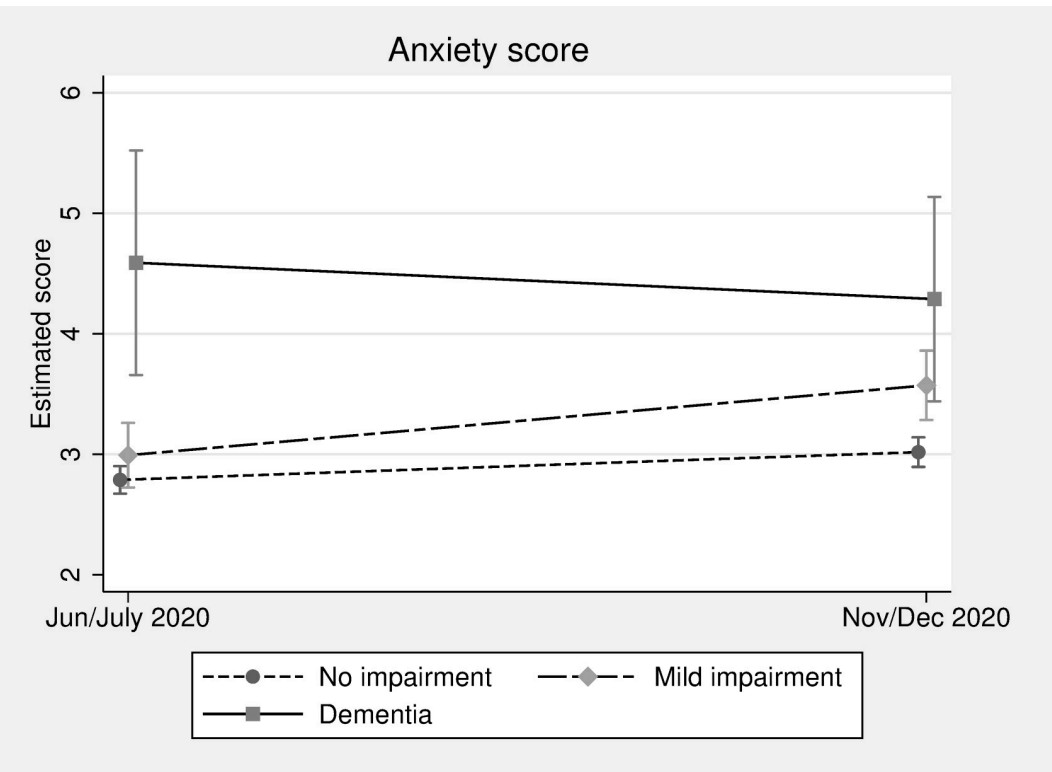

**Fig 2. Anxiety score (GAD-7) over time by cognitive function (estimated scores and 95% confidence intervals, adjusted for all available covariates).**

We did, however, identify some significant two-level interactions that provide further insights into the inequalities in mental health experienced by people with cognitive impairment (mild or dementia) during the pandemic. Models were subsequently estimated using all covariates featured in those reported above. We found significant interactions linking education to anxiety. For depression, we found significant results related to wealth and multi-morbidity (Fig 5).

Looking at wealth and depression, the results show that, prior to the pandemic, those with cognitive impairment in the poorest third of wealth had an estimated depression score of 1.62 (1.48 to 1.76), making them worse off compared to the other groups. This contrasts to those without impairment in the richest two-thirds, who scored 1.13 (1.07 to 1.19) on depression, positioning them better than either group with cognitive impairment; those with impairment in the richest two-thirds had an estimated depression score of 1.33 (1.20 to 1.45).

During the pandemic, however, we see notable changes, especially among those with no impairment in the poorest third of wealth, whose scores change from 1.25 (1.16 to 1.34) in 2018/2019 to 2.01 (1.91 to 2.12) in June/July. Their scores become much more similar to those with impairment also in the poorest third, who scored 2.03 (1.87 to 2.19) in June/July 2020. During the pandemic, the score for those with no impairment in the richest two-thirds was significantly lower than those for the poorest groups, at 1.76 (1.69 to 1.82) in June/July and 1.93 (1.85 to 2.01) in November/December. In other words, it appears that being in the poorest third of wealth is a stronger driver than cognitive function for estimated depression scores during the pandemic.

Turning to multimorbidity, the results illustrate that those with cognitive impairment and multimorbidity scored worse on depression than the groups without cognitive impairment

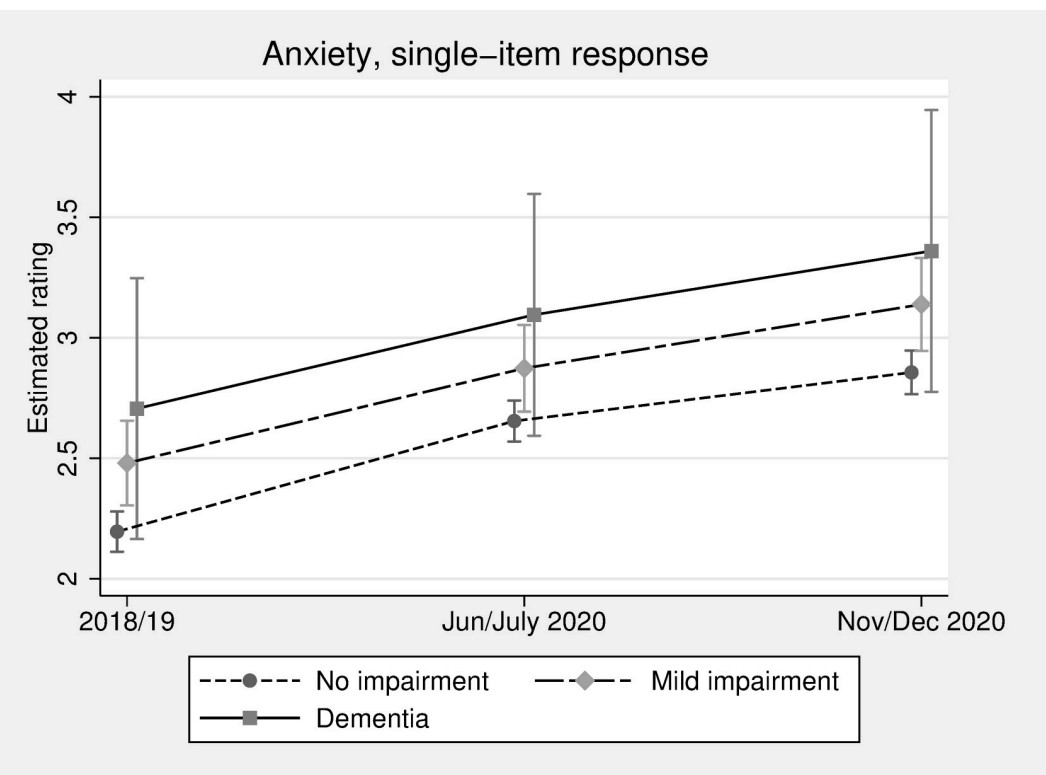

**Fig 3. Single-item anxiety rating over time by cognitive function (estimated ratings and 95% confidence intervals, adjusted for all available covariates).**

prior to the pandemic, with a score of 1.62 (1.46 to 1.77); this contrasts with a score of 1.24 (1.14 to 1.34) for those with multimorbidity but no impairment and 1.15 (1.09 to 1.20) for those without either multimorbidity or impairment. Just after the start of the pandemic, they were only slightly worse off than those with neither cognitive impairment nor multimorbidity, with scores increasing to 2.04 (1.87 to 2.22) and 1.80 (1.74 to 1.86), respectively. By November/ December, this difference had grown further, with scores of 2.35 (2.13 to 2.57) and 1.98 (1.90 to 2.06), respectively, but there were still no significant differences for those with only one of either cognitive impairment or multimorbidity.

Across the 2 assessments of the COVID-19 sub-study, there was a notable difference in anxiety score by education. In June/July, the only significant difference in anxiety score was between those with cognitive impairment and low education (3.28 (2.92 to 3.63)) and those with no impairment and high/medium education (2.73 (2.60 to 2.86)). This difference persisted by November/December, widening to 3.84 (3.47 to 4.21) and 3.01 (2.88 to 3.15), respectively, while those with no impairment and low education were also significantly lower in anxiety score, at 2.99 (2.73 to 3.25), than those with impairment and low education.

## Discussion

Using a representative sample living in private households in England, we have found that depression and anxiety worsened during the pandemic compared to before it for people with mild cognitive impairment or no impairment, whereas no statistically significant increases were found for those with dementia. We also found that differences according to cognitive function that existed before the pandemic were no longer present by June/July 2020, indicating

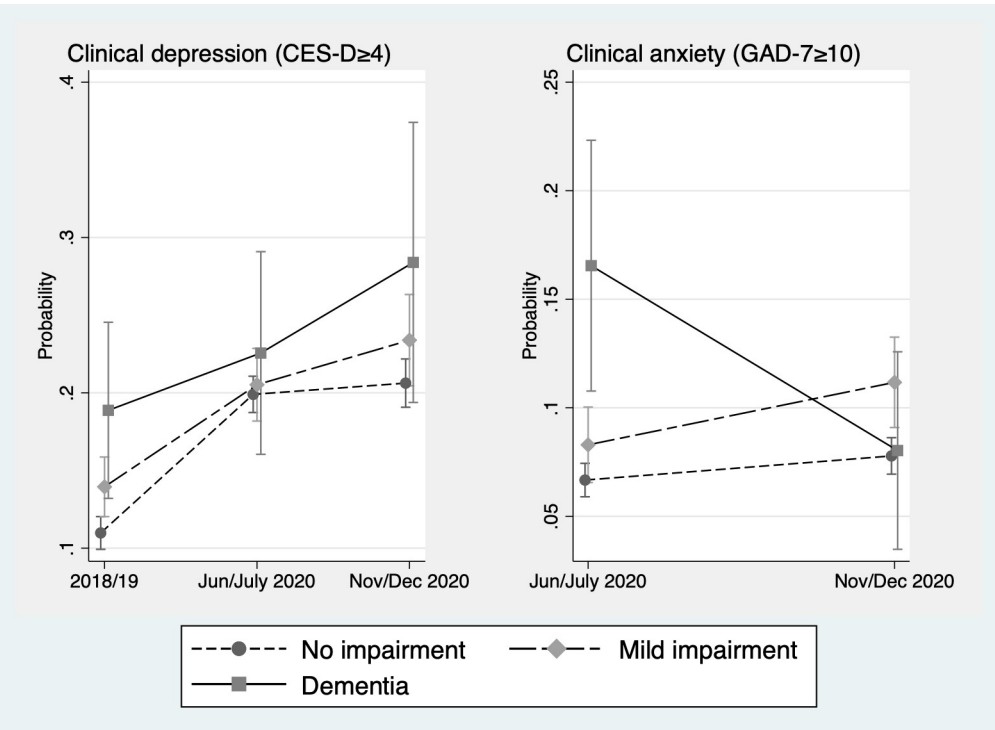

**Fig 4. Likely clinical depression and anxiety over time by cognitive function (estimated probabilities and 95% confidence intervals, adjusted for all available covariates).**

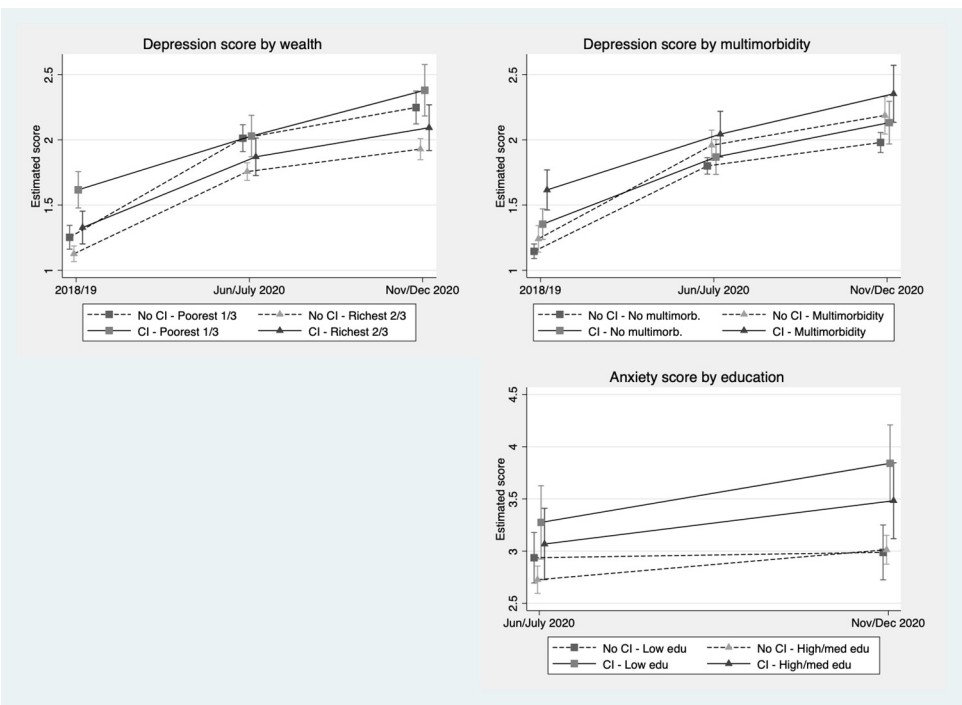

**Fig 5. Significant inequalities relating mental health and cognitive function (estimated scores and 95% confidence intervals, adjusted for all available covariates).**

a convergence in mental health across cognitive function groups during the pandemic. In November/December 2020, there also were no statistically significant differences among cognitive groups. Taken together, our findings provide partial support for our hypotheses on the relationship between mental health and cognitive impairment over the course of the pandemic.

Our multi-item scale of anxiety (GAD-7) provides insight only during the pandemic, as we unfortunately had no pre-pandemic comparable measure. With this measure, we did identify significant differences in anxiety between those with no impairment and those with dementia during the pandemic. The contrast with the results using the single-item measure may stem from an underlying distinction in the measured concepts, i.e., the single-item captures individual self-assessments of being anxious, whereas GAD-7 captures a broader perspective on generalized anxiety. Given potential concerns over the ability of respondents with cognitive impairment to provide reliable self-assessments, it may be that the results from a multiple-item measure like GAD-7 are more robust than a single response for identifying differences between cognitive groups.

With GAD-7, the mild impairment group also demonstrated significant change, being similar to the levels of those with no impairment in June/July 2020 but more similar to those with dementia by November/December 2020. This may relate to a deterioration in cognitive function among those with mild impairment during this time that stimulated increases in generalized anxiety. It may also suggest that those with mild impairment encountered anxiety-inducing challenges during the latter part of 2020 that other groups did not. In addition, although there was a significant difference according to cognitive function in November/December using GAD-7, the estimated scores do show a converging trend, possibly supporting the results from the single-item measure.

This convergence may also explain how our results sit against those from other research that found no change in anxiety during the pandemic among the adult UK population overall and lower levels for older age groups [15]. That research and others assessed anxiety looking at proportions with clinical scores on GAD-7 (i.e., $\geq 10$), which may attenuate the results and subsequent conclusions, partly explaining why findings related to the overall older population in England vary from ours accounting for cognitive function [21].

Another key insight from this work relates to the increase in likely clinical cases of depression for those with no impairment or mild cognitive impairment during the pandemic compared to 2018/2019. This highlights the potential increase in demand on mental health services that might be expected moving forward. Moreover, this sits alongside the finding that the level of likely clinical cases among those with dementia has not declined, so any challenges in service delivery that existed before the pandemic will likely persist.

Alongside these potential pressures on service delivery for mental health, we should also note that other research suggests that the negative impact of the pandemic on mental health was less acute for older adults compared to other age groups. This has been found in Scotland with respect to depression and anxiety [54] along with other UK-based studies [13,55]. It has been argued that this stems from a less pronounced difference in mental health among older adults comparing before and after the pandemic [56]. Our results may partially support this argument, at least with respect to recognizing little difference among those with dementia pre- and post-pandemic.

Our findings also stand in contrast to other work in the UK looking at mental health with respect to the pandemic. One study found that the negative impact on mental health extended through the initial months of the pandemic and started to improve from July 2020 [57], whereas another found a recovery trajectory in anxiety among UK adults from April [4]. Moreover, a large meta-analysis of longitudinal cohort studies found that the changes in

anxiety attributed to the pandemic appeared short-lived, with peaks appearing around March/April 2020 and declining by July, although there was still a small increase in depression between May and July [14].

Our data during the pandemic were collected in June/July and November/December 2020, yet we did not find a significant improvement between these time points. In fact, we found evidence for worsening mental health for those with no impairment or mild impairment. One possible explanation is the specific timeframes in which our data were collected. The assessment in June/July may have missed the initial spike in poor mental health reported in other studies. Our second assessment took place in November/December; this is later than those examined in the studies mentioned above. It also coincides with the second lockdown imposed in the UK in November, which was followed by easing and then further tightening of restrictions in December. These shifts may have had somewhat similar effects on mental health as the first lockdown. This may be further supported by evidence suggesting deterioration of mental health in the UK between July/August 2020 and February 2021 [15].

The differences in other results and ours may also relate to the various ways depression and anxiety have been assessed, for example, average scores on a scale, the proportions based on clinical cutoffs, or other measures used elsewhere like the 12-item General Health Questionnaire (GHQ-12). While there are alternative measures for depression and anxiety that have been used in other studies, CES-D has been used since the inception of ELSA as the main measure of depression and to facilitate international comparisons with other ageing cohort studies like the Health and Retirement Study based in the United States, while GAD-7 has been introduced for similar reasons. Both measures have also demonstrated good performance and are useful as self-reported assessments [58–61]. Moreover, our study focuses on cognitive impairment, making distinctions between dementia and mild cognitive impairment. Evidence related to these differences from longitudinal studies that include pre-pandemic assessments is limited; some studies suggest that changes in mental health observed among people with cognitive impairment were more related to expected changes linked to the impairment rather than the pandemic itself [38,40].

The analyses using the binary measures of likely clinical depression or anxiety give some insight into clinical relevance, particularly with respect to the result that the probability of likely clinical depression among those with no impairment or mild impairment was significantly higher during the pandemic than in 2018/2019. For the continuous measures, however, there is no consensus on what level of change in score constitutes clinical significance, although some work suggests, with respect to longer versions of CES-D, that somewhere between a 11% and 17% change across the scale would represent a minimal clinically important difference (MCID) [62,63]. With respect to CES-D, this would range 0.88 to 1.36, which suggests a clinically significant difference between 2018/2019 and November/December 2020 for those with no or mild impairment. One study of GAD-7 suggests an MCID of 4 points, a threshold not reached in our findings [64].

Across our analyses, this study has several strengths. Our data are drawn from a longitudinal sample of people aged 50+ in England, allowing us to examine outcomes at 3 distinct time points across the same people. Indeed, the COVID-19 sub-study achieved a notably high response rate for each wave of data collection and longitudinally. This strengthens our ability to generate robust results even during the public health crisis and social restrictions caused by the pandemic. We are also able to employ validated measures of depression and anxiety in addition to comparing the latter with a single-item measure to assess changes from before the pandemic. Finally, the breadth of the ELSA data allowed us to incorporate a wide range of adjustment variables to control for confounding in our models.

The strength of our findings is possibly limited by the relatively small number of people categorized with dementia. The wide confidence intervals this yielded may, in fact, hide true

differences that exist but are masked in the statistical results. Yet, our study also includes people categorized with mild cognitive impairment, strengthening the analytical perspective in relation to cognitive function. We also cannot know if people with dementia were less likely to respond to the survey despite high response rates overall. However, our study has much lower attrition between the 2 assessments during the pandemic than most of the other studies reported in 1 systematic review of mental health before and during the pandemic [14]. This could indicate that stability in mental health found elsewhere is influenced by the attrition of those with worse or declining mental health over time.

The implications for our findings can be generalized to people with cognitive impairment or dementia living in the community but not to those who live in care homes or other institutional settings. This implies that our findings do not apply to the 40% of older UK adults with dementia living in such places [65]. However, this is a strength of our study, as fewer studies of people with dementia have been conducted in a way to reflect those living in the general community. Moreover, despite any potential limitations, ELSA and its related sub-studies have been an invaluable resource for examining the experiences of older people before and during the pandemic, along with a more robust assessment of cognitive function, potentially being the best evidence for this to date.

Our findings highlight 2 key implications for future public health responses in England as we move through the post-pandemic phase. First, mental health services will need to be supported and adequately resourced to meet the predicted increased demand that will come from older adults, especially those not living with cognitive impairment or dementia. This is underscored by the potential demand from other age groups, who may have experienced worse impacts on mental health than what we have identified in our cohort.

Second, given we saw little significant change in mental health outcomes among those with dementia, we must recognize their need for support will continue to exist. In the near term at least, the challenges in delivering this support are less likely to relate to the cognitive impairment itself but to questions of accessibility and availability, especially if the supply of support is diverted to those with no impairment. Policymakers and care practitioners will, therefore, need to ensure that people with dementia have equal access to measures to support their mental health.

To conclude, this study has improved our understanding of the way that mental health changed for people with cognitive impairment and dementia due to the pandemic. We find evidence for increasing levels of depression and anxiety during the pandemic among those with mild cognitive impairment or no cognitive impairment, but not among those with dementia. Compared to before the pandemic, mental health has become more similar across cognitive function groups, suggesting a convergence that may impact future demand for support services.

## Supporting information

**S1 STROBE Checklist. STROBE checklist.**
(DOCX)

## Acknowledgments

The English Longitudinal Study of Ageing was developed by a team of researchers based at University College London, NatCen Social Research, the Institute for Fiscal Studies, the University of Manchester, and the University of East Anglia.

## Author Contributions

**Conceptualization:** Andrew Steptoe, Paola Zaninotto.

**Data curation:** Brian Beach.

**Formal analysis:** Brian Beach.

**Funding acquisition:** Andrew Steptoe, Paola Zaninotto.

**Investigation:** Brian Beach, Paola Zaninotto.

**Project administration:** Andrew Steptoe, Paola Zaninotto.

**Supervision:** Paola Zaninotto.

**Writing – original draft:** Brian Beach.

**Writing – review & editing:** Brian Beach, Andrew Steptoe, Paola Zaninotto.

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
