## [Editor Report · Decision Letter 0]

20 Dec 2022

Dear Dr Beach, 

Thank you for submitting your manuscript entitled "Changes in depression and anxiety among people with cognitive impairment and dementia during the COVID-19 pandemic: Analysis of the English Longitudinal Study of Ageing" for consideration by PLOS Medicine for our upcoming Special Issue.

Your manuscript has now been evaluated by the PLOS Medicine editorial staff as well as by an academic editor with relevant expertise and I am writing to let you know that we would like to send your submission out for external peer review.

Please re-submit your manuscript within two working days, i.e. by Dec 22 2022 11:59PM.

Kind regards,

Callam Davidson

Associate Editor

PLOS Medicine

---

## [Decision Letter · Decision Letter 1]

13 Feb 2023

Dear Dr. Beach,

Thank you very much for submitting your manuscript "Changes in depression and anxiety among people with cognitive impairment and dementia during the COVID-19 pandemic: Analysis of the English Longitudinal Study of Ageing" (PMEDICINE-D-22-04054R1) for consideration at PLOS Medicine for our upcoming Special Issue. 

Your paper was evaluated by an associate editor and discussed among all the editors here. It was also discussed with the guest editors of the Special Issue, and sent to independent reviewers, including a statistical reviewer. The reviews are appended at the bottom of this email and any accompanying reviewer attachments can be seen via the link below:

[LINK]

In light of these reviews, I am afraid that we will not be able to accept the manuscript for publication in the journal in its current form, but we would like to consider a revised version that addresses the reviewers' and editors' comments. Obviously we cannot make any decision about publication until we have seen the revised manuscript and your response, and we plan to seek re-review by one or more of the reviewers. 

We expect to receive your revised manuscript by Mar 06 2023 11:59PM. Please email us (plosmedicine@plos.org) if you have any questions or concerns.

We look forward to receiving your revised manuscript. 

Sincerely,

Callam Davidson, 

PLOS Medicine

plosmedicine.org

Please include continuous line numbering throughout your manuscript.

Abstract Methods and Findings:

* Please include the study design and number of participants.

* Please include the important covariates that are adjusted for in the analyses.

Please place citations in square brackets.

Please ensure that the study is reported according to the STROBE guideline, and include the completed STROBE checklist as Supporting Information. Please add the following statement, or similar, to the Methods: "This study is reported as per the Strengthening the Reporting of Observational Studies in Epidemiology (STROBE) guideline (S1 Checklist)."

Did your study have a prospective protocol or analysis plan? Please state this (either way) early in the Methods section.

For all observational studies, in the manuscript text, please indicate: (1) the specific hypotheses you intended to test, (2) the analytical methods by which you planned to test them, (3) the analyses you actually performed, and (4) when reported analyses differ from those that were planned, transparent explanations for differences that affect the reliability of the study's results. If a reported analysis was performed based on an interesting but unanticipated pattern in the data, please be clear that the analysis was data-driven.

Please include the numerators and denominators in Table 1.

Figures 1-5: Please begin the y-axes at zero or include a break in the axes. 

The terms gender and sex are not interchangeable (as discussed in https://www.who.int/health-topics/gender); please confirm that gender is the appropriate term for your study.

Some of the content in your Results would be better placed in the Discussion (any interpretation, e.g., ‘In other words, it appears that being in the poorest third…’).

Please consider tabulating your comparisons to facilitate interpretation for the reader.

Please remove funding details from your Acknowledgements and instead ensure they are included in your Financial Disclosure in the Submission Form.

Please remove the Author Contributions, Competing Interests, Data Availability and Disclaimer sections from the main text and ensure all relevant details are captured in the appropriate part of the Submission Form.

Please relocate the Ethical Approval Statement to the Methods section and confirm whether you sought advice from an Institutional Review Board. 

For Internet sources in the References, please include the date accessed.

Comments from the reviewers:

Reviewer #1: The authors propose a study exploring change in depression and anxiety between before and during the Covid-19 pandemic in three subsamples of people without cognitive impairment, with mild cognitive impairment, and with dementia. The phases in which data was collected and the large sample size make it possible to raise clarity over the results of previous studies on the topic. The analyses and results reported are very robust. I have some minor suggestions for the authors:

- In the introduction I would like to see some estimations on prevalence of depression and anxiety, MCI and dementia in the second half of life.

- Also, I would like to see discussion of how prevalence of depression and anxiety vary in late middle age, early old age, and advanced old age. I think a limitation of this study may be that all people aged 50+ very analyses together but different age groups may have different evolutions in terms of depression and anxiety.

- The study aim in the introduction is not very clear in terms of timepoints. Could the authors rephrase the below sentence and be more specific in terms of timepoints: "Using data from the English Longitudinal Study of Ageing (ELSA), we examine changes in depression and anxiety from before the pandemic across two time points during the pandemic with respect to three levels of cognitive function".

- In the introduction, when describing the study aims, I would spell out the sociodemographic inequalities investigated. 

- As the ELSA study started in 2002, where those who remained in the study in 2002 healthier than those who dropped out since 2002? I would comment this.

- The analyses section is very robust.

- How was multimorbidity defined? two or more conditions? Also which conditions were counted? Could the authors please explain this under Table 1 as a note

- For education could the authors explain what consists in low, medium, and high education under Table 1. 

- Minor comment: often the authors use "while" instead of "whereas".

- In the discussion, first paragraph, could the authors specify the year of Nov/Dec. The same in the remaining paragraphs - years should be specified to ease readability.

- Could the authors also discuss the size/clinical relevance of the changes found

- For those with mild impairment it would be interesting to discuss in the discussion whether the size of the change observed during the pandemic is comparable to the size of the change observed in individuals with mild cognitive impairment over the same period of time in non-pandemic contexts

Reviewer #2: The manuscript is a study about changes in depression and anxiety among people with cognitive impairment during COVID-19. I think the manuscript deserve publication in this journal however some changes are necessary.

- I think the N sample size needs to be in the abstract, the methods and Findings section and in Table 1.

- I would like to know how the authors handle the multiple comparisons made in the study and if they have made corrections for that. If not I would like to know the justification for that.

- In the result section it is only indicated the value in anxiety and depresion in differents temporal points (and CI) but not the value of change or a value about effect size with the CI or p value. I think this measure are needed. At least I would like to know why this measures are not reported.

Thank you for the opportunity to review the manuscript

Kind regards

Reviewer #3: This is a well-conducted study on the changes in depression and anxiety among people with cognitive impairment and dementia during the COVID-19 pandemic using the ELSA dataset. The study design, datasets, statistical methods and analyses, and presentation (tables and figures) and interpretation of the results are mostly adequate and of a good standard. However, still a few issues needing attention.

1) There is no mention of statistical methods at all in the abstract.

2) Sample size was not clearly mentioned. Can the authors please make the sample size clear at each survey throughout the manuscript including text, tables and figures?

3) Table 1. Count and percentage are both needed instead of just the percentage.

4) Can authors please confirm whether the estimates in Figure 1 to 5 were adjusted for all the confounders as described in the methods secion Page 13? 

5) Rating scales. There are so many rating scales for depression and anxiety, up to around 20 including the ones like HADS and etc. Can the authors go a bit further in the discussion on why and justifications of using CES-D or GAD-7 as compared to other widely used scales? Pros and cons?

6) In the abstract, it says "Using a clinical cutoff for likely depression (CES-D≥4), we found statistically significant increases in the probability of likely clinical depression between 2018/19 and Nov/Dec 2020 for those with no impairment (0.110 (0.099-0.120) to 0.206 (0.191-0.222)) and those with mild impairment (0.139 (0.120-0.159) to 0.234 (0.204-0.263))". I am not sure what is meant by 'probability of likely clinical depression'. By a cutoff, we then can identify a proportion of patients with likely depression, rather than probability, is that right?

[LINK]

---

## [Decision Letter · Decision Letter 2]

20 Mar 2023

Dear Dr. Beach,

Thank you very much for re-submitting your manuscript "Changes in depression and anxiety among people with cognitive impairment and dementia during the COVID-19 pandemic: Analysis of the English Longitudinal Study of Ageing" (PMEDICINE-D-22-04054R2) for review by PLOS Medicine.

I have discussed the paper with my colleagues and the academic editor and it was also seen again by two reviewers. I am pleased to say that provided the remaining editorial and production issues are dealt with we are planning to accept the paper for publication in the journal.

[LINK]

We hope to receive your revised manuscript within 4 working days. Please email us (plosmedicine@plos.org) if you have any questions or concerns.

We look forward to receiving the revised manuscript by Mar 24 2023 11:59PM.   

Sincerely,

Callam Davidson, 

Associate Editor 

PLOS Medicine

plosmedicine.org

Requests from Editors:

Please update your title to ‘Depression and anxiety in people with cognitive impairment and dementia during the COVID-19 pandemic: Analysis of the English Longitudinal Study of Ageing’.

‘All figures reported below reflect the full model, adjusted for all available covariates’ – please include this information in the legends of Figures 1-5. 

Author Summary: 

* Include the headline numbers from the study, such as the sample size and key findings under the question ‘What did the researchers do and find’. 

* Typo in bullet point 2 under the question ‘What do these findings mean’.

Please update the final paragraph of your introduction to reflect the hypotheses described at lines 97-104 (the abstract, author summary, and discussion ought to be similarly updated).

At line 202, the term "trend" is used to refer to a nonsignificant P value. The term trend should be used only when the test for trend has been conducted. Please revise accordingly.

I feel that the results relating to inequalities in mental health and cognitive function ought to be acknowledged, at least briefly, in the abstract and author summaries (this can be a single sentence summary). 

Reference 9 contains an asterisk, the meaning of which is unclear. 

Please check the panel labels in Figure 4 – it looks as though some symbols have been replaced with a question mark. 

Comments from Reviewers:

Reviewer #3: Thanks authors for their great effort to improve the manuscipt. I am satisfied with the response and revision. No further issues needing attention.

[LINK]

---

## [Editor Report · Decision Letter 3]

21 Mar 2023

Dear Dr. Beach,

Thank you very much for re-submitting your manuscript "Depression and anxiety in people with cognitive impairment and dementia during the COVID-19 pandemic: Analysis of the English Longitudinal Study of Ageing" (PMEDICINE-D-22-04054R3) for review by PLOS Medicine.

The remaining issues that need to be addressed are listed at the end of this email. We hope to receive your revised manuscript by EOB Friday 24 March. Please email us (plosmedicine@plos.org) if you have any questions or concerns.

Sincerely,

Callam Davidson, 

Associate Editor 

PLOS Medicine

plosmedicine.org

Requests from Editors:

I feel there is still some confusion around the presentation of the hypotheses. It is essential that the manuscript clearly and consistently describes the hypotheses you intended to test and the analytical methods used to test them.

The abstract and author summary both do not mention that the investigation was intended to investigate ‘how associations would vary across key sources of socioeconomic inequality’ (Methods), nor do they provide quantitative data to support the statement ‘Wealth and education appeared to be stronger drivers for depression and anxiety, respectively, than cognitive impairment’. 

Related to the above, I feel the description of this hypothesis in the Introduction (‘We also investigate whether key sociodemographic inequalities related to wealth, education, geographic region, and multimorbidity are linked to patterns in outcomes’) differs from that in the Methods. Please clarify exactly what hypothesis you intended to test and provide a harmonised description throughout the manuscript. 

Your study is observational and therefore causality cannot be inferred. Please adjust the way your hypotheses are worded to reflect this in the Abstract, Author Summary, and throughout. For example, I feel that the sentence ‘This study addresses such gaps in the evidence base to test whether depression and anxiety among older people was worse with increasing cognitive impairment and worsened from before to during the COVID-19 pandemic’ would be better presented as ‘This study investigated whether changes in depression and anxiety in older people during the COVID-19 pandemic were associated with level of cognitive impairment’.

---

## [Editor Report · Decision Letter 4]

27 Mar 2023

Dear Dr Beach, 

On behalf of my colleagues and the Guest Editor, Dr Lola Kola, I am pleased to inform you that we have agreed to publish your manuscript "Depression and anxiety in people with cognitive impairment and dementia during the COVID-19 pandemic: Analysis of the English Longitudinal Study of Ageing" (PMEDICINE-D-22-04054R4) in our upcoming Special Issue in PLOS Medicine.

When making the formatting changes, please also address the following editorial comment:

* Abstract: Please move the sentence in which you describe the limitations of your study from the end of the 'Abstract - Conclusions' section to the end of the 'Abstract - Methods and Findings' section (as it was in the previous draft). 

PRESS

We ask that you take this opportunity to read our Embargo Policy regarding the discussion, promotion and media coverage of work that is yet to be published by PLOS. As your manuscript is not yet published, it is bound by the conditions of our Embargo Policy. Please be aware that this policy is in place both to ensure that any press coverage of your article is fully substantiated and to provide a direct link between such coverage and the published work. For full details of our Embargo Policy, please visit http://www.plos.org/about/media-inquiries/embargo-policy/.

REPRODUCIBILITY

Sincerely, 

Callam Davidson 

Associate Editor 

PLOS Medicine